# Right Ventricle Mechanics and Function during Stress in Patients with Asymptomatic Primary Moderate to Severe Mitral Regurgitation and Preserved Left Ventricular Ejection Fraction

**DOI:** 10.3390/medicina56060303

**Published:** 2020-06-20

**Authors:** Rūta Žvirblytė, Ieva Merkytė, Eglė Tamulėnaitė, Agnė Saniukaitė, Vaida Mizarienė, Eglė Ereminienė, Jolanta Justina Vaškelytė

**Affiliations:** Department of Cardiology, Lithuanian University of Health Sciences, LT-44307 Kaunas, Lithuania; ieva.merkyte91@gmail.com (I.M.); egletamulenaite@gmail.com (E.T.); asaniukaite@gmail.com (A.S.); vaida.mizariene@lsmuni.lt (V.M.); egle.ereminiene@lsmuni.lt (E.E.); jolanta.vaskelyte@kaunoklinikos.lt (J.J.V.)

**Keywords:** primary mitral regurgitation, right ventricle function, stress echocardiography, speckle-tracking echocardiography

## Abstract

*Background and objectives*. Mitral regurgitation (MR) is usually dynamic and increasing with exertion. Stress may provoke symptoms, cause the progression of pulmonary hypertension (PH) and unmask subclinical changes of the left and right ventricle function. The aim of this study was to evaluate changes of right ventricle (RV) functional parameters during stress and to find out determinants of RV function in patients with MR. *Materials and methods*. We performed a prospective study that included patients with asymptomatic primary moderate to severe MR and preserved left ventricular (LV) ejection fraction (EF) at rest (≥60%). Conventional 2D echocardiography at rest and during stress (bicycle ergometry) and offline speckle tracking analysis were performed. *Results*. 80 patients were included as MR (50) and control (30) groups. Conventional functional and myocardial deformation parameters of RV were similar in both groups at all stages of exercise (*p* > 0.05). The grade of MR (*p* = 0.004) and higher LV global longitudinal strain (*p* = 0.037) contributed significantly to the changes of tricuspid annular plane systolic excursion (TAPSE) from rest to peak stress. Changes of MR ERA from the rest to peak stress were related to RV free wall longitudinal strain (FWLS) and four chambers longitudinal stain (4CLS) at rest (*p* = 0.011; *r* = −0.459 and *p* = 0.001; *r* = −0.572, respectively). Significant correlations between LV EF, stroke volume, cardiac output and RV fractional area change, S′, TAPSE, FWLS, 4CLS were obtained. However, systolic pulmonary artery pressure and RV functional, deformation parameters were not related (*p* > 0.05). *Conclusions*. Functional parameters of LV during exercise and severity of MR were significant determinants of RV function while PH has no correlation with it in patients with primary asymptomatic moderate to severe MR.

## 1. Introduction

Mitral regurgitation (MR) is one of the most common valvular heart diseases [1]. Severe MR leads to volume overload of the left ventricle (LV) and left atrium (LA), causes progression of postcapillary pulmonary hypertension (PH) and heart failure [2,3]. Due to mechanisms of compensation, patients could be asymptomatic for a long time even when MR is moderate or severe [4]; therefore, adequate treatment is quite often delayed and is provided too late.

The severity of MR is usually dynamic and increasing with exertion [5]. Stress may provoke symptoms, cause the progression of PH and unmask subclinical changes of the LV and right ventricle (RV) function [6]. Moreover, a decreased physical capacity, significantly increased severity of MR and increased PH (maximal pressure in RV more than 60 mmHg) during exercise are related to worse prognosis [5].

Secondary tricuspid regurgitation (TR) accompanying MR is associated with proportional changes of right-sided heart morphology and function [7,8]. PH before surgery is associated with more common postoperative failure of both RV and LV [9]. Dysfunction of RV is related to the negative prognostic impact in patients with MR. Therefore, evaluation of RV function and pulmonary artery pressure (PAP) during stress plays an important role in predicting outcomes and identifying worse prognosis and may help to optimise the management of patients with asymptomatic primary MR.

There is not enough data about the impact of RV mechanics, function and changes thereof during stress in patients with primary moderate to severe MR with preserved LV ejection fraction (EF).

Compared to LV, echocardiographic evaluation of RV is more challenging and limited [10,11]. Novel techniques such as three-dimensional echocardiography, speckle tracking echocardiography are promising and very useful for assessing accurate parameters of RV [12,13].

The aim of this study was to evaluate changes of RV functional parameters during stress and to find out determinants of RV function in patients with asymptomatic primary moderate to severe MR and preserved LV EF.

## 2. Materials and Methods

### 2.1. Study Population

We performed a prospective study that included patients with asymptomatic primary moderate to severe MR and preserved LV EF (≥60%) at rest.

A total of 88 patients older than 18 years were enrolled in the study. However, 8 of them were not suitable for off-line speckle tracking echocardiographic analysis, so they were excluded. The remaining 80 patients were included in the study. In total, 50 (62.5%) of them had moderate-severe MR (MR group), and 30 (37.5%) patients had no significant heart valve disease (control group). All the subjects gave their informed consent for inclusion before participating in the study. The study was conducted in accordance with the Declaration of Helsinki, and the protocol was approved by Kaunas Regional Biomedical Research Ethics Committee (Project identification code BE 2-54, 26 June 2018).

Patients were excluded from the study if they had:(1)Contraindications to stress testing or disability to perform bicycle ergometry(2)Clinically significant ischaemic heart disease (previous myocardial infarction, percutaneous coronary intervention, bypass surgery, present symptoms of angina pectoris or pathological stress test was obtained).(3)Reduced LV EF (less than 60% according to modified Simpson’s rule).(4)Oncological disease.(5)Asthma, chronic obstructive pulmonary disease or other lung disease.(6)Uncontrolled arterial hypertension.(7)Significant hypertrophy of LV (interventricular septal or LV posterior wall thickness >13 mm).(8)Previous cardiac surgery.(9)More than mild aortic valve damage or mitral stenosis.

We collected and analysed clinical data from medical documentation: age, gender, comorbidities, medications, symptoms of MR, heart failure or other diseases, risk factors of cardiovascular disease, data of objective investigation (weight, height, calculated body surface area and body mass index, heart rate, arterial blood pressure), electrocardiographic findings. Conventional 2D echocardiography at rest and during stress was performed in all patients.

### 2.2. Echocardiography

2D echocardiography at rest and during stress was performed by an experienced cardiologist. All echocardiographic scans were evaluated by the same investigator.

A 12-lead electrocardiogram (ECG) was taken throughout the test.

The conventional transthoracic 2D echocardiography system (EPIQ 7, Phillips Ultrasound, Inc., Bothell, WA, USA) with 1.5–4.6 MHz transducer was used. All measurements were obtained according to existing recommendations [13].

LV EF was calculated using modified Simpson’s biplane method (in the apical four- and two-chamber views LV endocardial borders at end-diastole and end-systole were manually traced) [13]. Stroke volume (SV) (difference between end diastolic and end systolic volume) was produced automatically using biplane Simpson’s method. Cardiac output (CO) calculated from SV and heart rate (using formula—SV × heart rate).

The function of RV was evaluated by measuring the tricuspid annular plane systolic excursion (TAPSE), the peak systolic velocity of tricuspid annulus (RV S′) and RV fractional area change (FAC). TAPSE was obtained using the M-mode in the apical four-chamber view, while RV S′ was obtained by tissue doppler imaging. FAC was calculated as the difference in end-diastolic and end-systolic area divided by end-diastolic area (also from apical four-chamber view by manually tracing the RV endocardial borders at end-diastole and end-systole) [13,14].

Severity of MR was graded using quantitative (effective regurgitation orifice area (ERA), regurgitation volume (RVol)) and qualitative criteria. ERA was obtained by evaluating proximal isovelocity surface area (PISA) method. RVol was derived from ERA and velocity time integral (VTI), which was obtained tracing continuous flow Doppler curve of MR [15,16].

Systolic PAP was estimated using RV systolic pressure added to a qualitative assessment of right atrial (RA) pressure. Systolic pressure of RV was derived from TR velocity (V) obtained at apical four chamber view using continuous Doppler by the Bernoulli equation (4 × TR V^2^ + 5 mm Hg assigned to RA pressure) [17,18]. Pressure of RA was derived according to measurements of the inferior vena cava dimensions during inspiration and expiration.

### 2.3. Exercise Stress Testing

A physical stress test (bicycle ergometry per protocol 25 W + 25 W every 3 min) was performed using a standard stress test protocol, monitoring blood pressure, heart rate, 12 lead ECG, clinical symptoms and signs at baseline, during stress and recovery phase (till the heart rate returned to the level it was at rest). Stress test was terminated prematurely in the presence of severe dyspnoea, chest pain or other intolerable symptoms, severe arrhythmia, more than 2 mm ST-segment elevation or depression, systolic blood pressure more than 230 mmHg, diastolic blood pressure more than 120 mmHg or a drop in systolic blood pressure more than 20 mmHg.

Parameters of LV and RV function, MR and TR severity, systolic PAP were evaluated at rest, during all stages of stress and during the recovery phase [5].

Maximal achieved workload was assessed by Watts and metabolic equivalents (METs).

### 2.4. Speckle Tracking Echocardiography 

Offline speckle tracking analysis (using Philips QLAB 13.0 program) was performed using images obtained at rest, during stress and recovery phase. Longitudinal strain and strain rate of LV were measured from apical four-, two- and three-chamber views according to existing guidelines [13]. Myocardial deformation parameters of RV (free wall longitudinal strain (FWLS) and RV 4 chamber longitudinal strain (4CLS)) were measured from apical four-chamber view [12]. Similarly to the LV, RV endocardial border (inner contour of RV myocardium) and epicardial border (outer contour of RV myocardium) were generated automatically (using “Auto RV” function) or manually traced and manually edited if necessary according to existing recommendations [12].

Cardiac cycles associated with ventricular or atrial extrasystolic beats were excluded from analysis. Also segments that had limited quality and/or poor tracking and were unsuitable for myocardial strain analysis were excluded.

### 2.5. Statistical Analysis

Statistical analyses were performed using SPSS 20.0 software. Data are presented as mean ± standard deviation (S.D) and as median (the first quartile-the third quartile) for continuous variables and as percentages for categorical variables. The Shapiro–Wilk test was used to determine the distribution of data in MR and control groups. Differences in the characteristics of the groups were assessed using independent-samples and paired samples *t*-tests (for normally distributed data) and Mann–Whitney tests (for non-normally distributed data) for continuous variables. Chi-squared tests were used to compare categorical variables. Pearson’s correlation coefficients were calculated to measure relations between two related variables. A multivariate stepwise regression analysis was performed to evaluate relationship between the increment of RV functional parameters during stress and parameters of MR severity, LV function and systolic PAP. The value of *p* < 0.05 was considered statistically significant.

## 3. Results

Based on MR presentation, 80 patients (mean age 60.20 ± 12.29 years; 57 (71.3%) female and 23 (28.7%) male) were included in the study as MR (*n*—50; 62.5%) and control (*n*—30; 37.5%) groups.

There were no significant differences in clinical characteristics between the MR and control groups (*p* > 0.05). The incidence of arterial hypertension, diabetes mellitus and other significant comorbidities also did not differ between the groups (*p* > 0.05). However, patients with MR had more episodes of previous paroxysmal arrhythmias (atrial fibrillation or flutter) (*p* = 0.015). Clinical characteristics of the study population are shown in Table 1.

The most common cause of MR in study sample was mitral prolapse (*n* = 24; 48%). In total, 11 patients (22%) with mitral prolapse had myxomatous degeneration of the valve. Other causes of MR were distributed as follows: 18 (36%) patients had degenerative mitral valve damage, 5 (10%) patients had rheumatic involvement of the valve, rupture of chordae was observed in 2 (4%) patients and isolated cleft of mitral valve was found in 1 (2%) patient.

At rest, MR ERA was 0.24 cm^2^ (0.16–0.31 cm^2^), RVol—41.48 mL (27–58 mL). MR ERA was increasing by 0.12 cm^2^ (0.06–0.15 cm^2^) from rest to peak stress.

Heart rate and blood pressure at rest, during the initial (25 W) stress, peak stress and during the recovery phase were similar in both groups (*p* > 0.05) (Figure 1).

The controls had a significantly better functional capacity. They achieved higher maximal workload, both Watts (*p* = 0.008) and METs (*p* = 0.003). However, only 12 (40%) controls and 15 (30%) patients with MR (*p* = 0.360) achieved submaximal heart rate according to their age. Higher S’ during peak stress was obtained in patients with higher achieved workload (by Watts−*r* = 0.355, *p* = 0.003 and by METs−*r* = 0.398, *p* = 0.002). Moreover, higher systolic PAP during peak stress was related to lower functional capacity (significant correlation with achieved Watts (*r* = −0.456, *p* < 0.001) and METs (*r* = −0.564, *p* < 0.001)).

Conventional parameters of RV function and myocardial deformation parameters of RV did not significantly differ between the groups at all stages of stress (*p* > 0.05) (Table 2).

Changes from rest to initial or peak stress of S′, TAPSE, FAC, RV FWLS and 4CLS (absolute value and percentage) during stress were similar in both groups (*p* > 0.05). However, parameters of longitudinal RV function (TAPSE, S′) were higher during initial (25 W), peak stress and recovery phase than at rest in patients with primary MR and in controls (Table 2).

In the MR group, changes of FAC (absolute value in %) and TAPSE (absolute value in mm) from rest to peak stress correlated to MR ERA (*p* = 0.035; *r* = 0.362 and *p* = 0.049; *r* = 0.317, respectively) and RVol (*p* = 0.023; *r* =0.452 and *p* = 0.031; *r* = 0.394, respectively) at rest. An increment of S′ (absolute value in cm/s) from initial to peak stress was significantly higher in patients with higher ERA at rest (*p* = 0.038; *r* = 0.333) and during stress (initial stress—*p* = 0.041; *r* = 0.353; peak stress—*p* = 0.011; *r* = 0.437) and higher RVol during peak stress (*p* = 0.033; *r* = 0.428).

The multivariate analysis (including MR ERA, RVol, grade, systolic PAP, LV EF and GLS at rest) showed that the grade of MR (*p* = 0.004) and higher LV GLS (*p* = 0.037) contributed significantly to the changes of TAPSE from rest to peak stress.

In patients with MR, changes of MR ERA (absolute value in cm^2^) from the rest to peak stress were significantly related to RV FWLS and 4CLS at rest (*p* = 0.011; *r* = −0.459 and *p* = 0.001; *r* = −0.572, respectively).

Higher systolic PAP during stress was noted in patients with MR (Figure 2). Moreover, systolic PAP from rest to peak stress significantly increased in MR group (*p* = 0.021). However, correlation between systolic PAP at all stages of exercise and parameters of RV function, deformation at rest or during stress was not obtained (*p* > 0.05).

In this study, 13 (26%) patients with MR had systolic PAP ≥60 mmHg during peak stress. However, conventional parameters of RV function and RV myocardial deformation parameters in those patients were similar as in others with MR.

LV EF and global longitudinal strain (GLS) rest were not significantly different between MR and control groups (Table 1). However, patients with MR had worse GLS during initial stress (−19.35 ± −3.29% vs. −21.30 ± −4.38%, *p* = 0.048) and tendency to lower GLS during peak stress (−19.28 ± −7.51% vs. −21.62 ± −315%, *p* = 0.286). In MR group, higher systolic PAP at rest significantly correlated with lower LV GLS at rest (*p* = 0.026, *r* = −0.352). LV GLS at rest and during stress was not related to parameters of RV function and deformation (*p* > 0.05). However, LV EF during initial (25 W) and peak stress correlated with RV FWLS (*p* = 0.027, *r* = −0.341 and *p* = 0.025, *r* = −0.354, respectively) and RV 4CLS (*p* = 0.018, *r* = −0.363 and *p* = 0.018, *r* = −0.369, respectively) during initial stress.

Subjects with the absence of LV contractile reserve had a tendency to lower S′ (15.62 ± 3.03 vs. 17.04 ± 2.78 cm/s; *p* = 0.066) and TAPSE (26.45 ± 4.14 vs. 27.63 ± 3.94 mm; *p* = 0.270) during peak stress (normal LV contractile reserve was defined as an exercise induced increase in LV EF ≥4% or in LV GLS ≥2% [19]. Patients with normal LV contractile reserve had better RV deformation parameters at all stages of exercise, although the difference was significant only for RV C4LS at rest (−28.24 ± 8.76% vs. −23.45 ± 8.36%; *p* = 0.049).

At rest, LV SV and CO were significantly higher in patients with MR than in controls (Table 1). In MR group myocardial deformation and conventional functional parameters of RV were related to LV SV and CO (Figure 3).

Correlations between: A—LV CO at rest and FAC during peak stress; B—LV CO at rest and RV S’ during peak stress; C—LV CO during peak stress and RV S′ during peak stress; D—LV CO during peak stress and FAC during peak stress; E—LV SV at rest and RV S′ during peak stress; F—LV SV during peak stress and RV FWLS.

## 4. Discussion

Our study evaluated changes of RV function during exercise stress and its predisposing factors in patients with asymptomatic primary moderate to severe MR. The main finding of our study was that the most important determinants of RV function during stress were functional parameters of LV (EF, SV, CO) and severity of MR while systolic PAP did not correlate with RV function.

RV function is closely related to symptom occurrence and exercise capacity in many clinical conditions. Echocardiographic evaluation of RV is still challenging and quite limited [10,11]. In every day clinical practice, TAPSE and S′ are the most commonly used, reproducible and validated indices of RV systolic function [20,21]. Several authors have proved that functional parameters of RV are increasing during exercise [22,23]. In present study, we also demonstrated that parameters of longitudinal RV function (TAPSE, S′) were higher during bicycle ergometry than at rest in patients with primary MR and in controls. However, at all stages of stress, all analysed RV functional parameters did not significantly differ between both groups. This could be related to similar (preserved) LV EF in all subjects and obtained significant correlation between RV functional parameters and LV EF.

We demonstrated that quantitative parameters of MR severity correlated with increment of RV function during exercise. In MR group changes of TAPSE, S′ and FAC during stress were significantly related to MR ERA and RVol at rest. During the past decade, the correlation between RV EF and MR ERA, RVol, regurgitant fraction was already noted in patients with severe chronic organic MR [24].

The role of MR in development of PH and the impact of PH on RV function have been established in previous studies [25,26,27]. When PH is present, chronic pressure overload of RV leads to cavity dilation and contractile dysfunction determining worse prognosis of patients with MR [27]. According to existing guidelines and recommendations, PH diagnosis should be verified by right heart catheterization [28,29]; however, transthoracic echocardiography helps to evaluate a level of PH probability and sometimes point out the cause of PH [29]. It is known that PAP is normally increasing during stress even in healthy population [30], but it should not exceed the upper limits, which is still difficult to estimate, especially during stress [31]. The elevated PAP during stress could be related to age, obesity, professional sports, various pathologic conditions and the most commonly it is due to left heart disease, valvular heart pathology [32]. In this study, we found higher systolic PAP during stress in patients with primary asymptomatic moderate to severe MR than in controls, while at rest, the difference was not significant. These results of our study correspond to findings from previous studies that chronic MR leads to development and progression of PH during stress [26,27].

Experts suggested systolic PAP ≥60 mmHg during exercise to be a significant threshold of negative prognostic value [5]. This was based on the fact that high systolic PAP is strongly related to elevated LV filling pressure resulting from exercise induced increase of MR severity. In our study, conventional parameters of RV function and RV myocardial deformation parameters were not dependent on the level of systolic PAP (*p* > 0.05). Similarly to our results, few previous studies have shown that impairment of RV function in patients with organic MR weakly depends on systolic PAP but mainly on LV remodeling and function [24,25]. In addition to this, we demonstrated that in MR group significantly higher S′, FAC and RV longitudinal strain during stress were obtained in patients with higher LV SV and CO.

Kusunose K. and colleagues presented that in patients with asymptomatic MR, RV strain at rest was associated with LV strain, other indices of RV function (FAC, rest and exercise TAPSE) and PAP; however, the only independent predictor of RV strain was resting LV strain [33]. Moreover, this study revealed that resting LV and RV strain, TAPSE during stress and exercise induced systolic PAP were significantly associated with the need for earlier mitral surgery [33]. Our study data showed that patients with moderate to severe MR and preserved LV EF had tendency to lower LV GSL during stress. Also, we found statistically significant correlation between higher systolic PAP and lower LV GLS at rest (*p* = 0.026). Furthermore, according to our results, myocardial deformation parameters of RV during stress were related to LV EF during stress. These findings are consistent with the statement of previous studies [25,33] that parameters of LV function are an important determinants of RV function.

Previous studies have demonstrated that LV contractile reserve (GLS increase ≤2% during stress) was a strong independent predictor of cardiac events in patients with ≥ moderate primary MR [19]. Our results showed that the absence of LV contractile reserve had tendency to worse RV function (lower TAPSE and S′) during peak stress and lower RV longitudinal strain during all stages of exercise. These findings confirm the trend that MR affects function of both ventricles. Several authors proved that biventricular impairment in patients with MR is a strong predictor of poor postoperative outcome and survival [24,25].

The data of present study suggest that in patients with primary asymptomatic moderate to severe MR, functional parameters of LV and severity of MR—not the level of PH—correlate to RV mechanics and function during stress. Further observational research and patients follow-up are necessary to evaluate long-term outcomes and prognostic value of these results.

## 5. Limitations

The main limitations of this study were the fact that 3D analysis of RV size, volume and function (RV EF and myocardial deformation parameters) and right heart catheterisation for PH evaluation were not performed. Moreover, the small sample size could be a relevant factor for some statistically insignificant results.

## 6. Conclusions

Conventional 2D echocardiography and speckle tracking echocardiography parameters did not detect any significant differences of RV function at rest and during stress in patients with and without MR. In patients with asymptomatic moderate to severe primary MR and preserved LV EF, systolic PAP during stress was higher and increased more than in controls. However, systolic PAP during stress did not correlate with parameters of RV function and deformation Functional parameters of LV (EF, contractile reserve, SV and CO) during exercise and parameters of MR severity were significant determinants of RV function while the level of PH has no correlation with it in patients with primary asymptomatic moderate to severe MR.

## Figures and Tables

**Figure 1 medicina-56-00303-f001:**
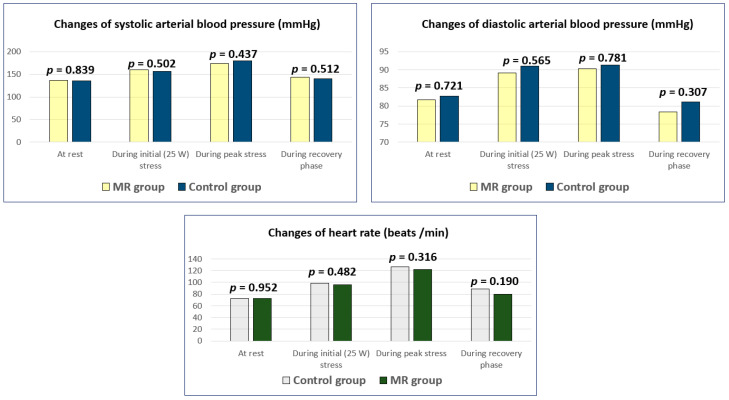
Changes of heart rate and arterial blood pressure during stress in MR and control groups. MR—Mitral regurgitation

**Figure 2 medicina-56-00303-f002:**
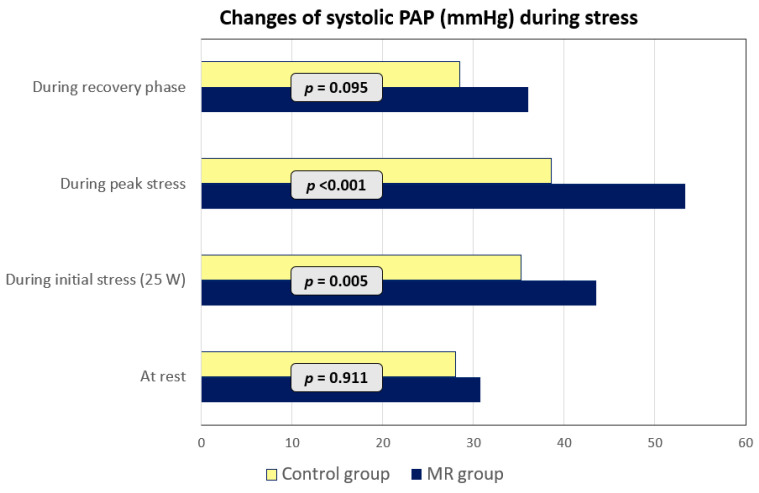
Changes of systolic PAP during stress. MR—mitral regurgitation; PAP—pulmonary artery pressure.

**Figure 3 medicina-56-00303-f003:**
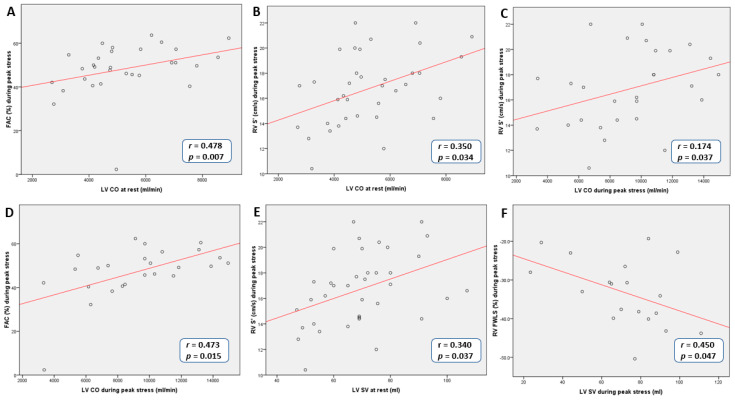
Correlations between LV SV, CO and parameters of RV function during peak stress in MR group. MR—mitral regurgitation, FAC—fractional area change, S′—peak systolic velocity of tricuspid annulus, RV—right ventricle, FWLS—free wall longitudinal strain, LV—left ventricle, SV—stroke volume, CO—cardiac output.

**Table 1 medicina-56-00303-t001:** Clinical characteristics and echocardiographic parameters of LV function at rest in MR and control groups.

Characteristics	MR Group (*n* = 50)	Control Group (*n* = 30)	*p* Value
Age (years)	61.88 ± 12.88	57.40 ± 10.89	0.115
Gender, male	13 (26%)	10 (33.33%)	0.611
Body surface area (m^2^)	1.81 ± 0.19	1.88 ± 0.17	0.384
Anamnesis of smoking	2 (4%)	1 (3.33%)	0.713
Arterial hypertension	31 (62%)	18 (60%)	0.690
Diabetes mellitus	2 (4%)	2 (6.67%)	0.400
Anamnesis of arrhythmias (atrial fibrillation/flutter)	14 (28%)	1 (3.33%)	0.015
Usage of β-blockers	32 (64%)	16 (53.33%)	0.629
Usage of ACE inhibitors/ARB	19 (38%)	11 (36.67%)	0.109
Usage of diuretics	15 (30%)	8 (26.67%)	0.094
LV EF (%) at rest	66.63 ± 4.65	67.28 ± 6.08	0.612
LV GLS (%) at rest	−18.17 ± −3.04	−17.91 ± −3.39	0.746
LV SV (mL) at rest	68.97 ± 19.11	57.89 ± 16.86	0.017
LV CO (L/min) at rest	5.18 ± 1.82	4.03± 1.09	0.005
Normal LV contractile reserve	28 (56%)	24 (80%)	0.048

MR—mitral regurgitation, ACE—angiotensin converting enzyme, ARB—angiotensin II receptor blocker, LV—left ventricular, EF—ejection fraction, GLS—global longitudinal strain, SV—stroke volume, CO—cardiac output. Values are means ± S.D. and N (%).

**Table 2 medicina-56-00303-t002:** Parameters of RV function at rest, during stress and recovery phase.

	MR Group (*n* = 50)	Control Group (*n* = 30)	*p* Value
**At Rest**			
FAC (%)	49.58 ± 8.43	50.43 ± 7.56	0.674
S′ (cm/s)	12.64 ± 1.96	12.52 ± 2.62	0.819
TAPSE (mm)	22.63 ± 3.18	22.01 ± 3.57	0.435
RV FWLS (%)	−28.64 ± −15.31	−29.57 ± −9.34	0.821
RV 4CLS (%)	−26.20 ± −9.19	−26.23± −7.89	0.990
**During Initial (25 W) Stress**			
FAC (%)	46.12 ± 8.33	46.79 ± 8.31	0.778
S′ (cm/s)	14.63 ± 2.48 *	14.10 ± 2.65 *	0.380
TAPSE (mm)	24.80 ± 4.36 **	25.48 ± 3.40 **	0.488
RV FWLS (%)	−30.86 ± −10.23	−33.26 ± −8.84	0.448
RV 4CLS (%)	−26.96 ± −8.08	−29.06 ± −8.99	0.433
**During Peak Stress**			
FAC (%)	48.79 ± 11.14	49.84 ± 10.92	0.764
S′ (cm/s)	16.37 ± 2.94 *	16.38 ± 3.26 *	0.983
TAPSE (mm)	27.35 ± 4.27 *	26.62 ± 4.39 *	0.491
RV FWLS (%)	−32.93 ± −8.63	−33.41 ± −7.61	0.895
RV 4CLS (%)	−27.74 ± −8.46	−26.96 ± −5.28	0.819
**During Recovery**			
FAC (%)	49.39 ± 8.99	45.27 ± 12.80	0.159
S′ (cm/s)	16.07 ± 3.79 *	15.80 ± 3.44 *	0.759
TAPSE (mm)	25.60 ± 3.97 *	25.22 ± 4.22 *	0.702
RV FWLS (%)	−34.83 ± −8.17	−33.88 ± −7.99	0.743
RV 4CLS (%)	−28.22 ± −7.22	−28.48 ± −5.22	0.920

MR—mitral regurgitation, FAC—fractional area change, S′—peak systolic velocity of tricuspid annulus, TAPSE—the tricuspid annular plane systolic excursion, RV—right ventricle, FWLS—free wall longitudinal strain, 4CLS—four chamber longitudinal strain. Values are means (SD). *—In the separate group the difference between the value of this parameter at rest and during this phase of stress was significant (*p* < 0.001); **—In the separate group the difference between the value of this parameter at rest and during this phase of stress was significant (*p* = 0.001).

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
