# Peer review of "Right Ventricle Mechanics and Function during Stress in Patients with Asymptomatic Primary Moderate to Severe Mitral Regurgitation and Preserved Left Ventricular Ejection Fraction"

_medicina, 2020, doi:10.3390/medicina56060303_

Round 1

Reviewer 1 Report

I had the pleasure of reviewing your interesting paper titled: "Right ventricle mechanics and function during stress in patients with asyntomatic primary moderate to severe mitral regurgitation and preserved left ventricular ejection franction".

The paper reads VERY well and cites most of the appropriate pieces of the literature.

The reviewer has some comments:

  1. Could you better specify whether those patients with more than mild aortic valve disease or mitral stenosis, rheumatic heart disease, hypertrophic obstructive cardiomyopathy,  history of MV surgery  or pulmonary diseases were excluded from the enrolled patients?
  2. Could you provide some information about the anatomical substrate of the regurgitation, possibly evaluated with transesophageal echo?
  3. Could you provide some information about functional capacity during exercise testing and any correlations with eco data?

    Kind regards

Author Response

Thank You for the review and comments. We added some more information and corrected the manuscript.

We would like to answer to Your comments:

  1. We can ensure that patients with more than mild aortic valve disease or mitral stenosis, hypertrophic obstructive cardiomyopathy, history of mitral valve surgery or pulmonary diseases were excluded from the study. 5 patients (10% of mitral regurgitation group) had rheumatic involvement of the valve however mitral stenosis was not prominent. Criteria of exclusion are corrected and better specified in the manuscript. (Lines 77-79 in the manuscript)
  2. The most common cause of MR in study sample was mitral prolapse (n=24; 48%). 11 patients (22%) with mitral prolapse had myxomatous degeneration of the valve. Other causes of MR were distributed as follows: 18 (36%) patients had degenerative mitral valve damage, 5 (10%) patients had rheumatic involvement of the valve, rupture of chordae was observed in 2 (4%) patients and isolated cleft of mitral valve was found in 1 (2%) patient. (Lines 158-162 in the manuscript). Transesophageal echocardiography (TEE) was performed in cases of severe MR, before surgery and in all cases of chordae’s ruptures and cleft. In this study we did not analyze views of TEE.
  3. Controls had significantly better functional capacity. They achieved higher maximal workload, both Watts (p=0.008) and METs (p=0.003). However, only 12 (40%) controls and 15 (30%) patients with MR (p=0.360) achieved submaximal heart rate according to their age. Higher S' during peak stress was obtained in patients with higher achieved workload (by Watts - r=0.355, p=0.003 and by METs - r=0.398, p=0.002). Also, higher systolic PAP during peak stress was related to lower functional capacity (to achieved Watts - r=-0.456, p<0.001; to METs - r=-0,564, p<0.001). (Lines 170-176 in the manuscript)

Reviewer 2 Report

Dear all,

it has been a pleasure to review this interesting and original manuscript.

The paper is well written, clear and easy to read. Bibliography is vast, articles well cited and clarify the content. The introduction is essential and focused correctly on the point of the question to address. Methods are correctly reported. Results are consistent and well described. Conclusion are consistent with the evidences and arguments are exhaustively presented.

Author Response

Thank You for the review.
